# Targeted gadofullerene for sensitive magnetic resonance imaging and risk-stratification of breast cancer

Zheng Han[1], Xiaohui Wu[1], Sarah Roelle[1], Chuheng Chen[1], William P. Schiemann[2] & Zheng-Rong Lu[1,2]

Molecular imaging of cancer biomarkers is critical for non-invasive accurate cancer detection and risk-stratification in precision healthcare. A peptide-targeted tri-gadolinium nitride metallofullerene, ZD2-Gd$_3$N@C80, is synthesised for sensitive molecular magnetic resonance imaging of extradomain-B fibronectin in aggressive tumours. ZD2-Gd$_3$N@C80 has superior $r_1$ and $r_2$ relaxivities of 223.8 and 344.7 mM$^{-1}$ s$^{-1}$ (1.5 T), respectively. It generates prominent contrast enhancement in aggressive MDA-MB-231 triple negative breast cancer in mice at a low dose (1.7 μmol kg$^{-1}$, 1 T), but not in oestrogen receptor-positive MCF-7 tumours. Strong tumour contrast enhancement is consistently observed in other triple negative breast cancer models, but not in low-risk slow-growing tumours. The dose of the contrast agent for effective molecular MRI is only slightly higher than that of ZD2-Cy5.5 (0.5 μmol kg$^{-1}$) in fluorescence imaging. These results demonstrate that high-sensitivity molecular magnetic resonance imaging with ZD2-Gd$_3$N@C80 may provide accurate detection and risk-stratification of high-risk tumours for precision healthcare of breast cancer.

[1] Case Center for Biomolecular Engineering, Department of Biomedical Engineering, Case Western Reserve University, 10900 Euclid Avenue, Cleveland, OH 44106, USA. [2] Case Comprehensive Cancer Center, Case Western Reserve University, Cleveland, OH 44106, USA. Zheng Han and Xiaohui Wu contributed equally to this work. Correspondence and requests for materials should be addressed to Z.-R.L. (email: zxl125@case.edu)

Precision medicine requires accurate detection and characterisation of tumours for tailoring personalised therapies to improve healthcare of cancer patients. Non-invasive sensitive imaging of tumour markers is essential for accurate cancer detection and characterisation of tumour aggressiveness[1]. Magnetic resonance imaging (MRI) is a powerful clinical imaging modality that provides high-resolution three-dimensional images of soft tissues. Magnetic resonance (MR) molecular imaging or molecular MRI of cancer biomarkers can facilitate non-invasive cancer detection and characterisation, image-guided interventions and therapeutic efficacy assessment in cancer precision medicine[2, 3]. High spatial resolution of MRI allows early detection of tumours as small as a few hundred cells, a few hundred microns in size[4]. Paramagnetic materials, including stable Gd(III) chelates and superparamagnetic iron oxide nanoparticles, are often used as contrast agents to increase $T_1$ and $T_2$ relaxation rates of water protons in the tissues of interest and to produce contrast enhancement for diagnostic imaging[5, 6]. Despite the continuous efforts in developing novel MRI contrast agents, there is a serious lack of safe and effective-targeted contrast agents for high-sensitivity molecular MRI in clinical practice.

MR signal enhancement is determined by the extent of increase in the water-proton relaxation rate, which is proportional to both the concentration and relaxivity of the contrast agents. Currently, the design of targeted MRI contrast agents mostly focuses on increasing their local concentration around the molecular targets, e.g. using nanoparticles with high payloads of paramagnetic materials[7]. Various nanosized-targeted contrast agents, including superparamagnetic iron oxide, Gd(III) and Mn(II)-containing nanoparticles[8–10], have shown promise in MR molecular imaging. However, because of their low relaxivity, a relatively large dose is required to produce detectable signal enhancement. High dose of nanoparticle contrast agents may cause unintended side effects due to their slow excretion and accumulation in normal tissues[11, 12]. Recently, a new strategy of targeting abundant biomarkers in the tumour microenvironment using small peptide conjugates of clinical Gd(III)-based MRI contrast agents has been developed[4]. Although this approach facilitates faster excretion of the targeted contrast agents in molecular MRI[13], a relatively high dose of these agents is still required to generate detectable signal enhancement.

Thus, the design and development of better contrast agents with high relaxivities is essential to significantly improve the sensitivity of molecular MRI and to reduce the local concentration and dose of the contrast agents, which will minimise potential dose-dependent toxic side effects. Recently, gadofullerenes have emerged as a novel class of paramagnetic materials with

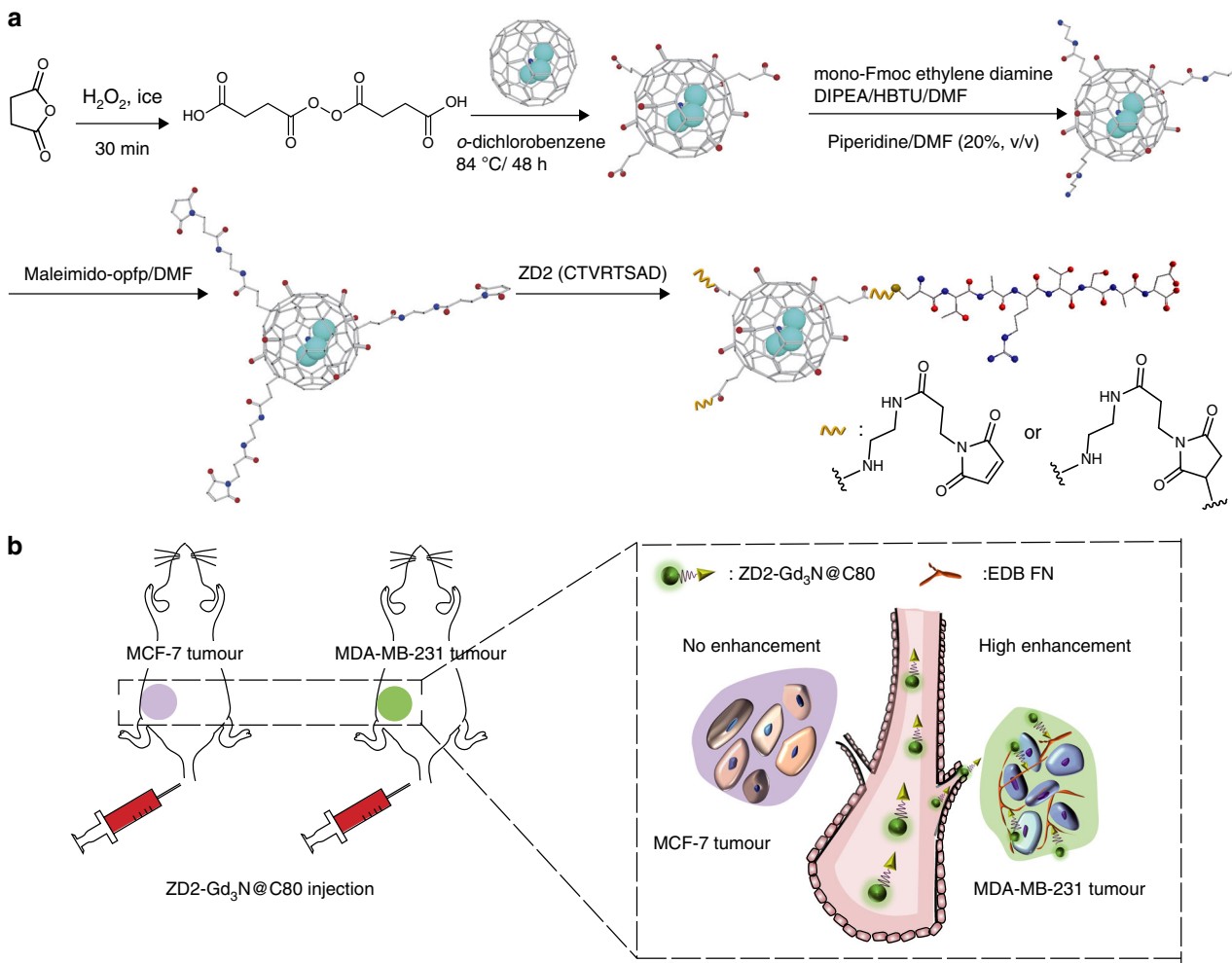

**Fig. 1** An EDB-FN-targeting gadofullerene for molecular MRI of breast cancer. **a** Schematic of synthesis of ZD2-Gd$_3$N@C80. *Cyan*, Gd; *blue*, nitrogen; *red*, oxygen; *grey*, hydrogen. **b** Illustration of tumour targeting with ZD2-Gd$_3$N@C80 for detection and characterisation of breast cancer in mouse models. MCF-7 and MDA-MB-231 cells were used to obtain low-risk and high-risk breast cancer xenografts, respectively. Intravenous injection of the EDB-FN-targeting agent, ZD2-Gd$_3$N@C80, results in different binding levels, corresponding to the EDB-FN expression and tumour aggressiveness, for tumour detection and characterisation with molecular MRI

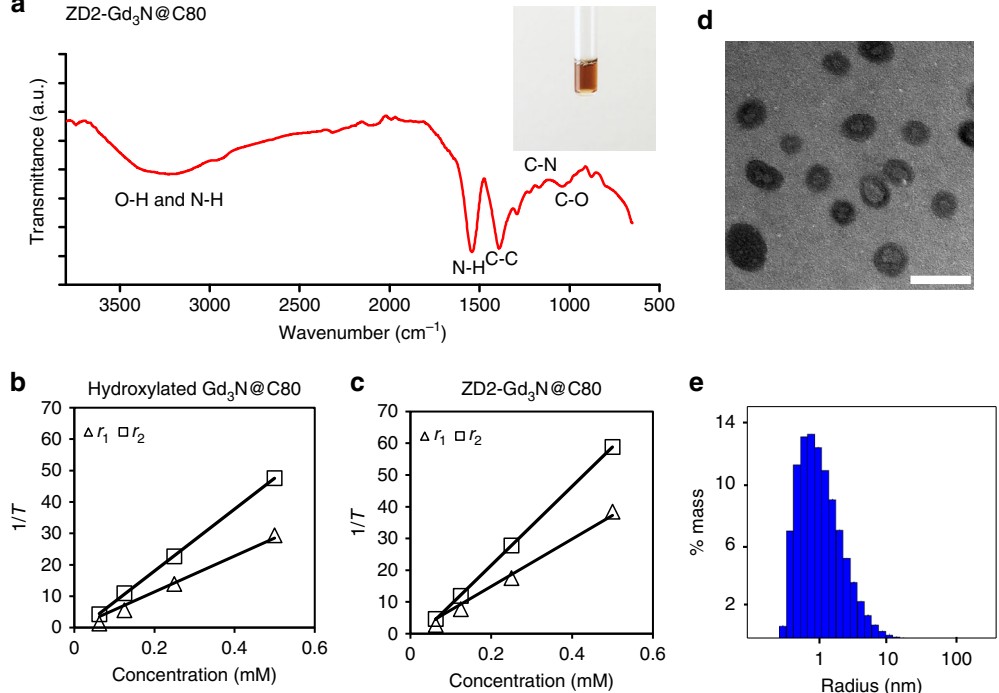

**Fig. 2** Characterisation of ZD2-Gd$_3$N@C80. **a** Fourier transform infrared spectroscopy (FTIR) of ZD2-Gd$_3$N@C80. *Inset*: photograph of ZD2-Gd$_3$N@C80 aqueous solution (0.083 mM). Plots of $1/T_1$ and $1/T_2$ vs. contrast agent concentrations for calculation of $r_1$ and $r_2$ relaxivities of **b** hydroxylated Gd$_3$N@C80 ($r_1 = 171.3$ mM$^{-1}$ s$^{-1}$; $r_2 = 295.5$ mM$^{-1}$ s$^{-1}$) and **c** ZD2-Gd$_3$N@C80 ($r_1 = 223.8$ mM$^{-1}$ s$^{-1}$ and $r_2 = 344.7$ mM$^{-1}$ s$^{-1}$) at 1.5 Tesla. **d** TEM images (*scale bar* 5 nm) and **e** DLS size distribution of ZD2-Gd$_3$N@C80 (average radius: 1.40 nm; polydispersity: 132%)

superior relaxivities, suitable for the design of targeted MRI contrast agents for sensitive molecular MRI[14–17]. For example, hydroxylated tri-gadolinium nitride metallofullerene Gd$_3$N@C80 possesses an $r_1$ relaxivity that is ~20 times that of conventional Gd-based contrast agents (GBCAs)[5, 18]. In addition, Gd(III) ions are encapsulated in the fullerene cage, preventing the release of toxic free Gd(III) ions into the body[19]. The small size (ca. 1 nm) of gadofullerenes also allows for complete clearance from the body via renal filtration. Therefore, targeted gadofullerenes are promising targeted MRI contrast agents that can address the limitations and toxicity of the existing MRI contrast agents for safe and sensitive molecular MRI in clinical practice.

Extradomain-B fibronectin (EDB-FN) is a marker for epithelial-to-mesenchymal transition (EMT), a biological process associated with tumour invasion, metastasis and drug resistance[20–23]. EDB-FN is highly expressed in the extracellular matrix of many types of aggressive human cancers[24, 25]. Clinical evidence shows that EDB-FN overexpression is associated with poor prognosis of a variety of cancers[26–28]. Thus, it is a promising target for cancer detection and characterisation with molecular MRI. We have identified a small peptide ZD2 (Cys-Thr-Val-Arg-Thr-Ser-Ala-Asp) for specific targeting to EDB-FN[24]. We have shown that Gd(HP-DO3A) modified with linear ZD2 peptide (Thr-Val-Arg-Thr-Ser-Ala-Asp) can be used to characterise prostate cancer aggressiveness in MRI[29].

In this work, we synthesise a high-relaxivity-targeted contrast agent by conjugating a small peptide ZD2 to hydroxylated Gd$_3$N@C80, ZD2-Gd$_3$N@C80, for sensitive molecular MRI of breast cancer. The targeted contrast agent has a superior $T_1$ relaxivity, about 20 times higher than the conventional Gd-based MRI contrast agents. The effectiveness of the agent for sensitive detection of aggressive tumours and risk-stratification is tested in multiple aggressive triple negative breast cancer (TNBC) and low-risk breast cancer models in mice. MRI with the targeted

contrast agent at significantly reduced doses produces strong signal enhancement in aggressive TNBC tumours, not in slow-growing low-risk breast tumours. The targeted contrast agent ZD2-Gd$_3$N@C80 is effective for sensitive molecular MRI for the detection and risk-stratification of aggressive breast cancer.

## Results

Gd$_3$N@C80 was first oxidised with succinic acid peroxide and NaOH to introduce carboxyl and hydroxyl groups on the fullerene cage surface (Fig. 1)[30]. MALDI-TOF mass spectrometric analysis of the hydroxylated Gd$_3$N@C80 suggested an estimated structure of Gd$_3$N@C80(OH)$_{18}$(CH$_2$CH$_2$COOH)$_6$. Some of the carboxyl groups in the hydroxylated Gd$_3$N@C80 were then converted into amines, followed by reaction with maleimido-opfp, yielding maleimido-Gd$_3$N@C80 for conjugation with thiol-bearing ZD2 peptide (Fig. 1). An excess of ZD2 peptide was used to conjugate to maleimido-Gd$_3$N@C80. The N–H and C–N peaks in Fourier transform infrared spectroscopy (FTIR) (Fig. 2a) indicated successful conjugation of the ZD2 peptide. The structure was also characterised with MALDI-TOF mass spectrometry, indicating approximately one ZD2 peptide was conjugated to each Gd$_3$N@C80.

The peptide-targeted tri-gadolinium nitride metallofullerene ZD2-Gd$_3$N@C80 showed complete water solubility (Fig. 2a), which is essential for further clinical development. The $r_1$ and $r_2$ relaxivities of hydroxylated Gd$_3$N@C80 and ZD2-Gd$_3$@NC80 were determined at 1.5 Tesla. The hydroxylated Gd$_3$N@C80 had $r_1$ and $r_2$ relaxivities of 57.1 and 98.5 mM$^{-1}$ s$^{-1}$ per Gd ion(III), respectively (Fig. 2b). ZD2-Gd$_3$@NC80 had superior $r_1$ and $r_2$ relaxivities of 223.8 and 344.7 mM$^{-1}$ s$^{-1}$ per molecule or 74.6 and 114.9 mM$^{-1}$ s$^{-1}$ per Gd(III) ion (Fig. 2c), respectively. The increased relaxivities of ZD2-Gd$_3$@NC80 may be attributed to slower tumbling rate and increased rotational correlation time of

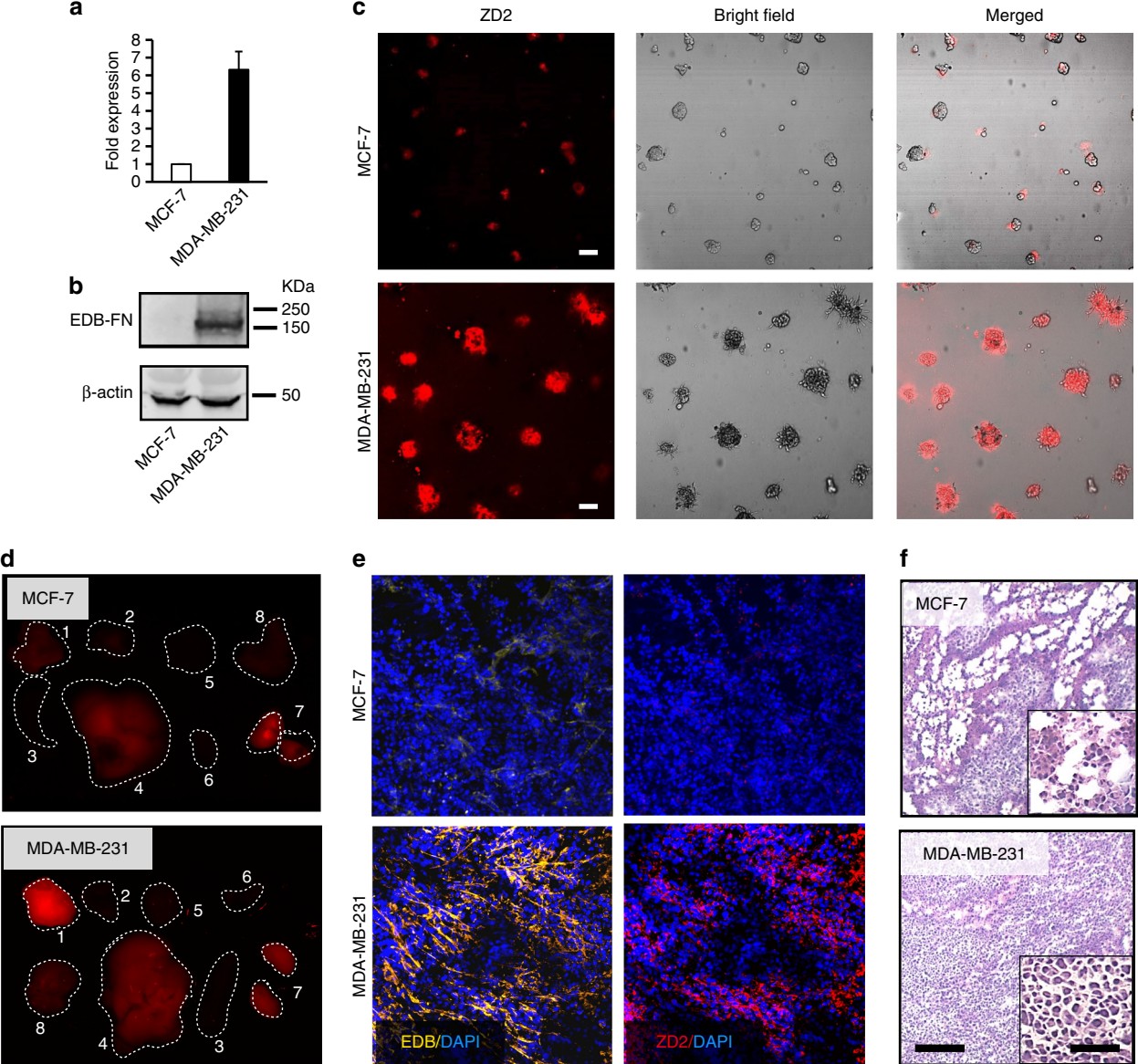

**Fig. 3** EDB-FN overexpression is a signature of aggressive breast cancer. **a** RT-PCR analysis of EDB-FN mRNA levels in MCF-7 and MDA-MB-231 cells, showing increased EDB-FN expression in MDA-MB-231 cells (data are presented as mean ± s.e.m. $n = 3$; two-tailed $t$-test: $P < 0.05$). The data are presented as ratios to the mRNA level of MCF-7. The mRNA level of β-actin was used for normalisation. **b** Western blot analysis of EDB-FN expression in MCF-7 and MDA-MB-231 tumours. β-actin was used as a loading control. **c** Representative fluorescence images of ZD2-Cy5.5 (*red*) binding, bright field images of 3D culture of MCF-7 and MDA-MB-231 cells, and the overlay of fluorescence images with bright field images. *Scale bar* 50 μm. **d** Ex vivo fluorescence images of tumour and organs collected from mice bearing MCF-7 and MDA-MB-231 tumours at 3 h after injection of 10 nmol ZD2-Cy5.5. *Numbers* in the images indicate the following tissues: 1 tumour; 2 muscle; 3 spleen; 4 liver; 5 brain; 6 heart; 7 kidney; 8 lung. **e** Analysis of EDB-FN expression and ZD2-Cy5.5 binding in MCF-7 and MDA-MB-231 tumour sections. DAPI was used for staining nuclei. Pseudo-colours in the confocal images are assigned as follows: *red*, ZD2 peptide; *green*, BC-1; and *blue*, nucleus. *Scale bar* 25 μm. **f** H&E staining showing the morphology of MCF-7 and MDA-MB-231 tumour sections. *Scale bar* 50 μm. *Inset*: enlarged images of the tumour sections (*scale bar* 10 μm)

the targeted agent due to the increased size after peptide conjugation. The $r_1$ relaxivity of ZD2-Gd$_3$@NC80 is almost 20 times that of clinical Gd(III)-based contrast agents, including Gd-DTPA and Gd(HP-DO3A)[5]. The high $r_1$ relaxivity of the hydroxylated Gd$_3$N@C80 is attributed to strong magnetisation of the hydroxyl protons on the cage by the encapsulated Gd (III) ions and rapid exchange rate of the protons with water protons in the surrounding bulk. This superior relaxivity is critical to improving the sensitivity of contrast enhanced MRI, especially $T_1$-weighted MRI, for molecular imaging at low doses on most clinical scanners with relatively low magnetic field strengths (1.5 and 3 Tesla). At 7 Tesla, the $r_1$ relaxivities of

hydroxylated Gd$_3$N@C80 and ZD2-Gd$_3$N@C80 were 24.68 and 24.78 mM$^{-1}$ s$^{-1}$ per Gd(III) ion, respectively (Supplementary Fig. 1). It appears that the slower tumbling rate and increased rotational correlation time of ZD2-Gd$_3$N@C80 had less effect on improving the relaxivities at the high magnetic field strength. ZD2-Gd$_3$N@C80 had an average diameter of 2.8 nm, as determined by transmission electron microscopy (TEM) (Fig. 2d) and dynamic light scattering (DLS) (Fig. 2e), which is smaller than the renal filtration threshold. This small size is necessary for rapid extravasation, target binding for effective molecular MRI and the elimination of the unbound agent from systemic circulation via renal filtration.

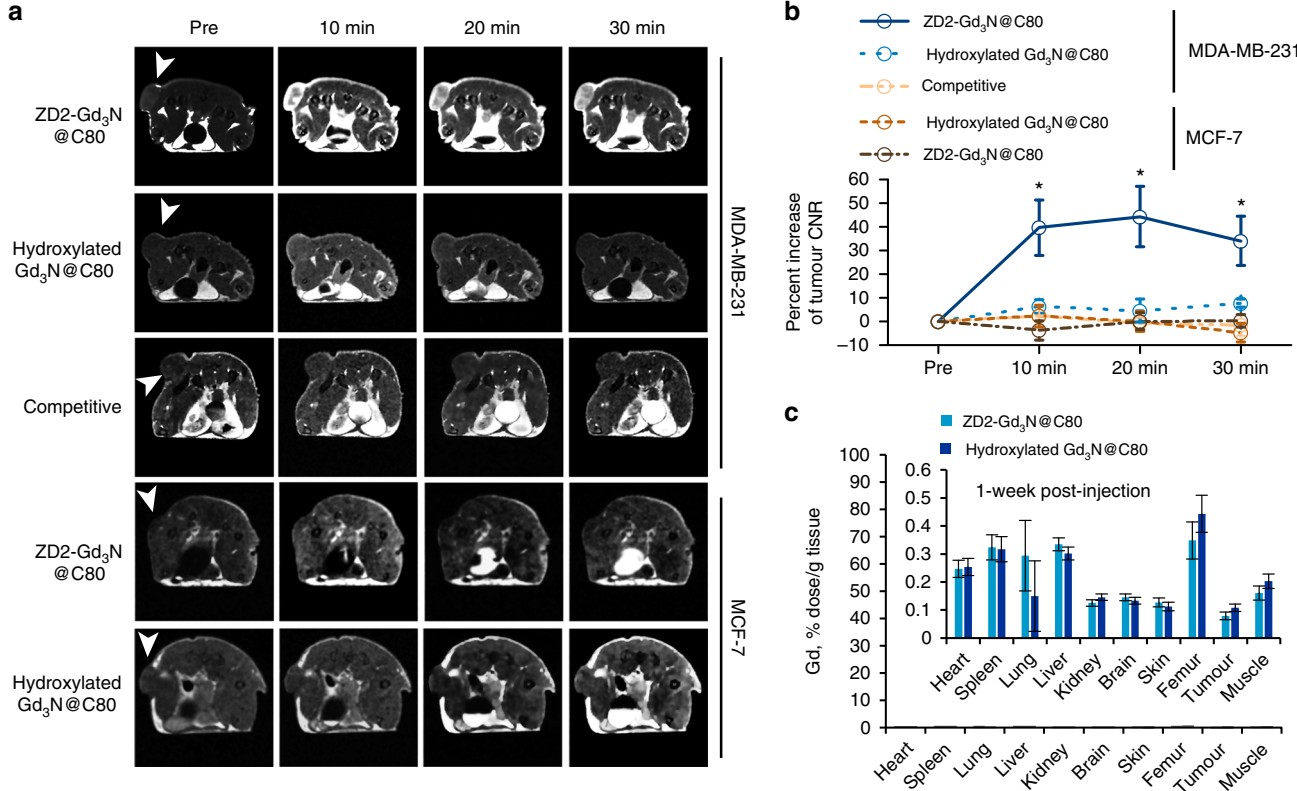

**Fig. 4** Contrast enhanced MRI with ZD2-Gd$_3$N@C80 of MDA-MB-231 and MCF-7 tumours in mice. **a** Representative axial $T_1$-weighted 2D spin-echo MRI images of MDA-MB-231 and MCF-7 tumours in mice. Images were acquired before and at 10, 20 and 30 min after injection of ZD2-Gd$_3$N@C80 and hydroxylated Gd$_3$N@C80 at a dose of 1.67 μmol, or a mixture of 25 μmol kg$^{-1}$ free ZD2 and 1.67 μmol ZD2-Gd$_3$N@C80 (competitive group). Tumour locations are indicated by *white arrow heads*. **b** Analysis of percentage increase of tumour contrast-to-noise ratio (*CNR*) from images acquired in groups indicated in (**a**) (data are presented as mean ± s.e.m. $n = 4$ for MDA-MB-231 tumours and $n = 3$ for MCF-7 tumours. *$P < 0.05$ for comparison of the increased CNR ratio of ZD2-Gd$_3$N@C80 in MDA-MB-231 group with that in all the other groups). **c** Gd biodistribution at 1 week after injection of ZD2-Gd$_3$N@C80 or hydroxylated Gd$_3$N@C80 in MDA-MB-231 tumour models. There was no statistical difference between retention of the contrast agents in all the tested tissues (data are presented as mean ± s.e.m. $n = 3$)

EDB-FN expression was determined in three TNBC cell lines (MDA-MB-231, Hs578T and BT549) and three oestrogen receptor (ER)-positive breast cancer cell lines (MCF-7, ZR-75-1 and T47D) using quantitative real-time polymerase chain reaction (qRT-PCR). The mRNA levels of EDB-FN in all the TNBC lines were significantly higher than those in the ER-positive cell lines (Fig. 3a and Supplementary Fig. 2). Western blot analysis also revealed abundant expression of EDB-FN in MDA-MB-231 tumours and negligible expression in MCF-7 tumours (Fig. 3b). In matrigel-based three-dimensional (3D) culture, MDA-MB-231 cells were able to form large spherical structures, indicating the aggressive nature of the TNBC cells (Fig. 3c). In contrast, MCF-7 cells formed smaller cell clusters with limited matrigel invasion, suggesting the low invasiveness of the cells (Fig. 3c). The MDA-MB-231 spheres incubated with ZD2-Cy5.5 showed strong fluorescence intensity under confocal microscopy, indicating high expression of EDB-FN protein and strong binding of the peptide probe in these spheres. In comparison, MCF-7 cells showed lower ZD2-Cy5.5 binding and weaker fluorescence intensity (Fig. 3c).

To further validate the in vivo targeting specificity of the ZD2 peptide, ZD2-Cy5.5 was intravenously injected at a dose of 0.5 μmol kg$^{-1}$ into the MDA-MB-231 and MCF-7 tumour-bearing mice. Figure 3d shows the fluorescence images of ZD-Cy5.5 in the tumours and other major organs at 3 h after injection. Fluorescence intensity was greater in the MDA-MB-231 tumours than the MCF-7 tumours and normal tissues and organs.

Relatively high fluorescence signal intensity was seen in the liver and kidneys, because the probe is mainly excreted via these organs. EDB-FN expression and ZD2-Cy5.5 binding in the two tumour models were also verified by correlating immunofluorescence staining of EDB-FN and Cy5.5 fluorescence imaging (Fig. 3e). MDA-MB-231 tumour sections were rich in EDB-FN expression and showed strong Cy5.5 fluorescence, whereas little EDB-FN expression and ZD2-Cy5.5 binding were seen in the MCF-7 tumour sections. Microscopically, the aggressive MDA-MB-231 TNBC tumour sections displayed a higher cell density than the ER-positive MCF-7 tumours (Fig. 3f). Taken together, these results validate the strong positive correlation of EDB-FN expression with tumour aggressiveness and the specific binding of ZD2 peptide to the highly abundant EDB-FN in the MDA-MB-231 tumours.

We next tested whether molecular MRI with ZD2-Gd$_3$N@C80 at 1 Tesla could detect the aggressive MDA-MB-231 tumours and differentiate the aggressive TNBC tumours from MCF-7 tumours in animal models. MR image acquisition was performed with mice bearing MDA-MB-231 and MCF-7 tumour xenografts before and after intravenous injection of ZD2-Gd$_3$N@C80 at a dose of 1.67 μmol kg$^{-1}$ or 5 μmol-Gd/kg, which is 20 times less than the standard dose of conventional clinical contrast agents, such as Gd-DTPA and Gd(HP-DO3A)[31]. Significant signal enhancement was observed in the MDA-MB-231 tumours for at least 30 min after the injection, while little enhancement was observed in the MCF-7 tumours or both tumours injected with non-targeted hydroxylated

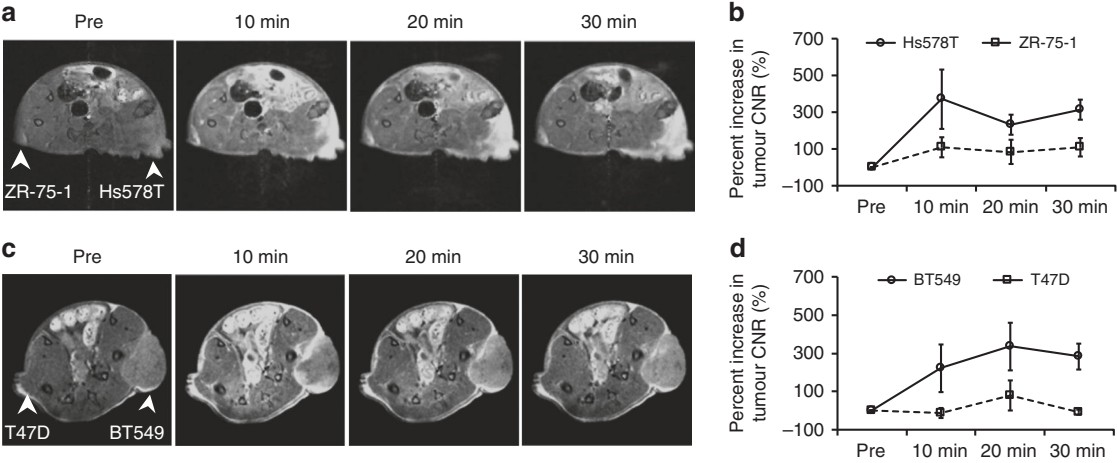

**Fig. 5** Molecular MRI with ZD2-Gd$_3$N@C80 of other breast tumours in mice. Representative axial MR images (7 Tesla) and tumour CNR analysis of mice bearing Hs578T and ZR-75-1 (**a**, **b**), BT549 and T47D (**c**, **d**) xenografts in flanks. Images were acquired at pre-injection and 30 min post-injection of 20 μmol Gd/kg ZD2-Gd$_3$N@C80 (data are presented as mean ± s.e.m. $n = 3$, two-tailed $t$-test; *$P < 0.05$). Tumour locations are indicated with *white arrows*

Gd$_3$N@C80 (Fig. 4a, b). Co-injection of 1.67 μmol kg$^{-1}$ of ZD2-Gd$_3$N@C80 with 25 μmol kg$^{-1}$ of ZD2 peptide significantly reduced the signal enhancement in MDA-MB-231 tumours due to competitive binding to the target. Quantitative analysis revealed that ZD2-Gd$_3$N@C80 produced 39 to 45% increase of contrast-to-noise ratio (CNR) in the MDA-MB-231 tumours (Fig. 4b). The smaller size of ZD2-Gd$_3$N@C80 facilitates rapid tumour extravasation, target binding, and clearance from circulation, and enables rapid sensitive detection of aggressive TNBC with high CNR. No significant difference was observed in signal enhancement in the normal tissues before and after contrast injection in all the tested groups, except for the excretory organs: kidneys and bladder (Supplementary Fig. 3). Increased signal intensity in the bladder indicates the excretion of unbound ZD2-Gd$_3$N@C80 via renal filtration.

The systemic retention of both ZD2-Gd$_3$N@C80 and hydroxylated Gd$_3$N@C80 was determined at 1 week post-injection using ICP-OES. The amount of Gd(III) in the body was near the detection limit of the ICP-OES. Both the agents had <0.5% of the injected dose remaining in the tissues and organs at 1 week post-injection (Fig. 4c). The potential release of the free Gd(III) ions was also tested by in vitro transmetallation assay with human serum. No transmetallation of ZD2-Gd$_3$N@C80 (<0.12%) was observed (Supplementary Fig. 4). Studies show that fullerene cages are highly stable and the metal ions are stably enclosed in these cages[18, 30]. Metallofullerene cages have been reported to even withstand exposure to β-radiation[18]. The residual endogenous metal ions seen in our study could be the result of the weak complexation of the peptide with the endogenous metal ions. Consistent with previous reports[19, 32], the entrapment of the toxic Gd(III) ions in the fullerene cage prevents their systemic release and tissue interaction, which is critical for the safety of the Gd(III)-based contrast agents in consideration of clinical translation. These results suggest that ZD2-Gd$_3$N@C80 has the potential to overcome the reported toxic side effects of the existing GBCAs, caused by the release and retention of free Gd(III) ions in the body.

The potential of ZD2-Gd$_3$N@C80 for differentiating breast cancer aggressiveness was further assessed in mice bearing ZR-75-1, Hs578T, T47D and BT549 breast cancer xenografts. The ER-positive ZR-75-1 and T47D xenografts showed a very slow rate of growth and relatively smaller tumour sizes as compared with the Hs578T and BT549 TNBC tumours. At 7 T, injection of 20 μmol Gd/kg ZD2-Gd$_3$N@C80 produced greater signal enhancement and significantly higher CNR increase in the fast-growing Hs578T and

BT549 tumours than in the slow-growing ZR-75-1 and T47D tumours (Fig. 5). These results demonstrate that ZD2-Gd$_3$N@C80 is an effective high-relaxivity-targeted contrast agent that can improve the sensitivity of molecular MRI for the detection and characterisation of aggressive breast tumours.

This study has demonstrated that the peptide-targeted gadofullerene with superior relaxivities can significantly improve the sensitivity of molecular MRI in detection and characterisation of aggressive breast cancer. We have shown that the superior relaxivity of ZD2-Gd$_3$N@C80 markedly improved the sensitivity of MR imaging of the oncoprotein EDB-FN in aggressive breast cancer. MRI with ZD2-Gd$_3$N@C80 produces robust signal enhancement in aggressive MDA-MB-231 TNBC tumours at a much lower dose (1.67 μmol kg$^{-1}$) at 1 Tesla, a sensitivity comparable with that of fluorescence imaging of ZD2-Cy5.5 (0.5 μmol kg$^{-1}$) in Fig. 3. The targeted contrast agent also produced strong enhancement in two other TNBC tumour models, not in slow-growing low-risk tumours at a high-field strength (7 Tesla) and reduced dose. These results have demonstrated the ability of the novel-targeted contrast agent for differentiating the fast-growing aggressive TNBC tumours from the slow-growing, non-metastatic and ER-positive breast cancers. Our work pioneers effective non-invasive differentiation between breast tumours of different aggressiveness using contrast enhanced MRI. EDB-FN is also highly expressed in other types of aggressive tumours, including prostate cancer[24], head and neck cancer[33] and ovarian cancer[34]. Therefore, molecular MRI with ZD2-Gd$_3$N@C80 can potentially be used for accurate detection and characterisation of a broad spectrum of aggressive cancers with high sensitivity and superior resolution. The low dose, rapid renal clearance and absence of potential release of free Gd(III) ions of ZD2-Gd$_3$N@C80 are advantageous safety features for improving the safety profile of GBCAs in clinical use. Clinical translation of molecular MRI with ZD2-Gd$_3$N@C80 has the potential to overcome the limitations of current imaging technologies and to significantly improve the accuracy of early detection and characterisation of high-risk breast cancer for precision healthcare of cancer patients.

## Methods

**Materials**. Succinic anhydride and *o*-dichlorobenzene were purchased from Sigma-Aldrich (St. Louis, MO, USA). Mono-Fmoc ethylenediamine was purchased from Combi-Blocks (San Diego, CA, USA). N,N,N′,N′-tetramethyl-*O*-(1H-benzotriazol-1-yl)uronium hexafluorophosphate, *O*-(benzotriazol-1-yl)-N,N,N′,N′-tetramethyluronium hexafluorophosphate (HBTU) and all protected amino acids for

peptide synthesis were purchased from Anaspec (Fremont, CA, USA). *N,N*-Diisopropylethylamine (DIPEA) was purchased from MP Biomedicals, LLS (Solon, OH, USA. Dimethylformamide (DMF), dichloromethane (DCM) and piperidine were purchased from Fisher Scientific (Pittsburgh, PA, USA). Gd$_3$N@C80 was purchased from SES Research (Houston, TX, USA). Maleimide-opfp (pentafluorophenyl 3-(2,5-dioxo-2,5-dihydro-1H-pyrrol-1-yl)propanoate) was synthesised according to a reported protocol[13].

**Oxidation of Gd$_3$N@C80.** At 0 °C, succinic anhydride (1.5 g, 15 mmol) was added dropwise to hydrogen peroxide and the mixture was stirred for 30 min. The product was washed with pure water and filtered, followed by lyophilisation for further use. Succinic acid acyl peroxide (8.0 mg, 5 equiv) was added to Gd$_3$N@C80 (10 mg) in 10 mL o-dichlorobenzene. The resultant solution was de-aerated by flashing with nitrogen and heated at 84 °C for 48 h. Additional succinic acid acyl peroxide (8 mg each time) was added at an interval of 12 h during the reaction. A brown sludge precipitated from the solution at the end of the reaction. Then, 20 mL 0.2 M NaOH aqueous solution was added to extract the water-soluble product. The top aqueous layer was deep brown and the bottom layer was colourless. The top layer was concentrated, adjusted to pH of 3–4 and purified with a PD10 column, and lyophilised to yield Gd$_3$N@C80(OH)$_{18}$(CH$_2$CH$_2$COOH)$_6$ (**2**) (yield, 74%). MALDI-TOF ($m/z$, [M+Na]$^+$): 2221(obsd); 2212 (calc).

**Synthesis of ZD2-Gd$_3$N@C80.** Gd$_3$N@C80(OH)$_{18}$(CH$_2$CH$_2$COOH)$_6$ (6 mg) and mono-Fmoc ethylenediamine (5 equiv) were dissolved in DMF, and then HBTU (5 equiv) and DIPEA (5 equiv) were added. The reaction was stirred at room temperature for 2 h. Piperidine/DMF (20%, v/v) was used to remove the protecting group of Fmoc. Afterwards, the product was precipitated in cold ether to obtain Gd$_3$N@C80(OH)$_{18}$(CH$_2$CH$_2$COOH)$_x$(NH$_2$)$_y$ (**3**) of brown colour. (yield, 63%). Gd$_3$N@C80(OH)$_{18}$(CH$_2$CH$_2$COOH)$_x$(NH$_2$)$_y$ (3 mg) was dissolved in DMF (5 mL) and excess maleimido-opfp (20 mg) was added. The reaction continued for 30 min before precipitating in cold ether to give maleimido-containing hydroxylated Gd$_3$N@C80. ZD2-Cys peptide (sequence: Cys-Thr-Val-Arg-Thr-Ser-Ala-Asp) was synthesised in solid phase using standard Fmoc-chemistry. Excess ZD2-Cys and Gd$_3$N@C80(OH)$_{18}$(CH$_2$CH$_2$COOH)$_x$(MAL)$_y$ were dissolved in pure water and stirred for 30 min. Then the solution was concentrated and purified with PD10 column. The final product ZD2-Gd$_3$N@C80 was collected and lyophilised (yield, 82%). MALDI-TOF ([M+2Na+2K+H]$^+$: $m/z$ = 3927 (obsd); 3922 [estimated for Gd$_3$N@C80(OH)$_{18}$(COOH)$_2$(MAL)$_4$-Cys-ZD2].

**Synthesis of ZD2-Cy5.5.** ZD2 peptide (sequence: Thr-Val-Arg-Thr-Ser-Ala-Asp) was synthesised in solid phase using standard Fmoc-chemistry. After Fmoc removal with 10% piperidine, Fmoc-9-amino-4,7-dioxanonanoic acid (ChemPep, Inc., Wellington, FL, USA) was added to the peptide sequence. After Fmoc removal with 10% piperidine, the resin was washed with DMF/DCM and air-dried, and 10 mg of the dried resin was swelled in DCM for 1 h, followed by reaction with 3 mg Cy5.5-NHS ester (Lumiprobe Corporation, Hallandale Beach, FL, USA) in presence of 5 µL DIPEA. Reaction was stirred overnight at room temperature. Excess Cy5.5-NHS ester was removed by filtration and washing with DMF/DCM 10 mL three times. Peptides were cleaved off resin using TIPS, and precipitated in cold ether. The products were separated from ether by centrifugation at 4000×*g*. The final product was characterised by MALDI-TOF mass spectrometry ([M+1]$^+$: $m/z$ = 1473.76 (obsd); 1472.78 (calc.)). The product was lyophilised and reconstituted in 500 µL PBS. The concentration of the solution was characterised by measuring absorbance at 650 nm.

**FTIR and relaxivity measurement.** Infrared spectrum of ZD2-Gd3N@C80 was performed using the Cary 630 FTIR Spectrometer (Agilent Technologies, Santa Clara, CA, USA). Lyophilised hydroxylated Gd$_3$N@C80 or ZD2-Gd$_3$N@C80 was reconstituted to a serial dilution in water. The solutions were pipetted in NMR tubes (500 µL in each tube), and placed in a relaxometer (Bruker) at 1.5 Tesla. For relaxivity measurement at 7 Tesla, NMR tubes containing contrast agent solutions were bundled and placed in a mouse coil in a horizontal 7 Tesla Bruker scanner (Bruker Biospin Co., Billerica, MA). $T_1$ maps of the solutions were acquired using a previously reported method[29]. $T_1$ and $T_2$ values of each solution were measured. The $r_1$ and $r_2$ relaxivities of the contrast agent were calculated as the slope of the plot of $1/T_1$ and $1/T_2$ relaxation rates against the concentrations.

**Cell culture.** MDA-MB-231, Hs578T, BT549, MCF-7, ZR-75-1 and T47D were acquired from American Type Culture Collection (ATCC, Rockville, MD, USA). MDA-MB-231 cells were maintained in Dulbecco Modified Eagle Medium (DMEM) supplemented with 10% fetal bovine serum (FBS) and 5% penicillin/streptomycin (pen/strep). Hs578T, BT549 and MCF-7 cells were maintained in Eagle' Minimum Essential Medium (EMEM) supplemented with 0.01 mg mL$^{-1}$ human recombinant insulin, 10% FBS and 5% pen/strep. ZR-75-1 and T47D were maintained in RPMI 1640 medium supplemented with 10% FBS and 5% pen/strep. Three-dimensional culture of cells was achieved by an 'on-top' matrigel method reported previously[35]. Briefly, on a glass-bottom plate a thick layer of matrigel was coated, followed by plating MDA-MB-231 or MCF-7 cells on top of the matrigel. Once 3D sphere or clusters formed, ZD2-Cy5.5 was added to the medium to the

final concentration of 250 nM. Binding of ZD2-Cy5.5 to the 3D spheres was evaluated using a confocal laser scanning microscope (Olympus Corporation, Tokyo, Japan) after culturing for 1 h.

**RT-PCR.** RNA was harvested from cells using the RNeasy Plus Mini Kit (Cat. # 74134, Qiagen, Hamburg, Germany) and reverse transcription was carried out with the miScript II RT Kit (Cat. # 218161, Qiagen, Hamburg, Germany), according to manufacturer's protocols. qRT-PCR was performed using SYBR Green PCR Master Mix (Cat. # 4309155, Applied BioSystems, Foster City, CA, USA) and the Eppendorf RealPlex Thermocycler (Eppendorf, Hauppauge, NY, USA). Cycle threshold (Ct) values were evaluated by the RealPlex 2.2 software system (Eppendorf, Hauppauge, NY, USA). Expression levels of human EDB-FN were analysed in triplicate by the $2^{-\Delta\Delta Ct}$ method and normalised to the expression levels of β-actin. Significance was found when $P \leq 0.05$. Primers were obtained from Invitrogen (Carlsbad, CA, USA) and sequences are as follows: human EDB-FN forward: 5′-CCTGGAGTACAATGTCAGTG-3′, human EDB-FN reverse: 5′-GGTGGAGCCCAGGTGACA-3′, human β-actin forward: 5′-GTTGTCGAC-GACGAGCG-3′, human β-actin reverse: 5′-AGCACAGAGCCTCGCCTTT-3′.

**Animal tumour models.** Female athymic nude mice were purchased from the Case Comprehensive Cancer Center and housed in the Case Center Imaging Research. All animal experiments were carried out according to a protocol approved by the IACUC of Case Western Reserve University. To initiate tumour xenografts, cells cultured in 150 mm dishes were trypsinised and centrifuged. Cell pellets from each dish were suspended in 100 µL PBS and reconstituted with Corning Matrigel Matrix High Concentration (Corning, Corning, NY, USA) to a concentration of $2 \times 10^7$ cells per mL on ice. The suspended cells in 100 µL matrigel solution were injected subcutaneously in the flank of mice (4–6 weeks) using a 19-gauge needle. After injection, a plug in flank was formed due to matrigel gelling. Tumours were allowed to grow for at least a month before imaging. To prepare mice bearing Hs578T on the right flank and ZR-75-1 on the left flank or BT549 on the right flank and T47D on the left flank), ZR-75-1 and T47D cell inoculations were performed at 4 weeks prior to Hs578T and BT549 inoculations. MRI were performed at 5 weeks after inoculating Hs578T and BT549 cells.

**Western blot.** The MDA-MB-231 and MCF-7 tumours were collected and homogenised in 200–500 µL T-PER buffer (Thermo Fisher Scientific) supplemented with the protease inhibitor cocktail (Sigma-Aldrich) and PMSF (phenylmethanesulfonyl fluoride) (Sigma-Aldrich). Centrifugation at 10,000×*g* for 10 min at 4 °C was used to remove insoluble components. Protein concentrations were quantified by BCA assay (Biorad, Hercules, CA, USA). Proteins of 25 µg were loaded for SDS-PAGE and transferred to polyvinylidene difluoride (PVDF) membranes (Invitrogen, Carlsbad, CA, USA). The anti-EDB-FN antibody (ab154210, BC-1, Abcam, Hercules, CA) and fluorescein-conjugated anti-mouse secondary antibody (ab97264, Abcam) were used. The fluorescein-conjugated anti-β-actin antibody was used for visualisation of β-actin. The Typhoon trio scanner (GE healthcare) was used for visualisation of EDB-FN and β-actin bands using the channel for fluorescein.

**Fluorescence imaging.** To determine the distribution of ZD2-Cy5.5 in the major organs and tumours, 10 nmol ZD2-Cy5.5 was injected in tumour-bearing mice through the tail vein. At 3 h after injection, mice were killed. Tumours and organs were collected and imaged with CSi Maestro imaging system (Woburn, MA, USA) using the deep red filter sets (exposure time: 1000 ms).

**Histological analysis.** Tumours were embedded in Optimum Cutting Tempera-ture Compound (OCT) and kept froze in −80 °C. Tumour tissues sectioned at the thickness of 5 µm were then fixed and permeabilised with cold acetone. BSA (1%) in PBS was used to block the tissue at room temperature for 1 h. Mouse anti-EDB-FN antibody, BC-1 (ab154210, Abcam, Cambridge, MA, USA), diluted with 1% BSA (1:500) was applied to the tissue and incubated at room temperature for 1 h. After extensive washing, AlexFluor594-conjugated anti-mouse IgG antibody (1:500) (Invitrogen, Eugene, OR, USA) was applied and incubated with the tissue for 1 h. Tissue sections were then counterstained with Prolong Gold antifade mounting medium with DAPI (Invitrogen). Confocal laser scanning microscopy (FV1000, Olympus, Japan) was used to acquire the fluorescence images of the tissues. H&E staining of the tissue was performed at the tissue resource core of Case Western Reserve University. H&E staining images were acquired using the Virtual slide microscope VS120 (Olympus).

**MR imaging.** MR images of the mice bearing MDA-MB-231 or MCF-7 tumour xenografts were acquired on an Aspect M3 small animal MRI scanner (1 Tesla). Mice were placed on a holder with the temperature maintained at 37°C, with isoflurane/oxygen mixture supplied to the mice through a nose cone. A thin catheter filled with PBS was connected to the tail vein of the mice. After mice were placed in the coil, a pilot scan was performed to adjust mice to the proper location in the coil. An axial $T_1$-weighted sequence (TR = 500 ms; TE = 9 ms; flip angle = 90°; field of view (FOV) = 3 cm × 3 cm; matrix size = 128 × 128 × 8;

slice thickness = 2 mm; interslice distance = 1 mm) was then used to acquire the images of the tumours before and 10 min, 20 min and 30 min after contrast injection. Images were exported into DICOM data, which were then processed and analysed using Matlab (Natick, MA, USA). The CNR ratio of tumours in the images was calculated as the difference between tumour mean intensity minus muscle mean intensity, divided by the noise. Three-dimensional images of mice were acquired using a gradient echo $T_1$-weighted sequence with the following parameters: TR = 17 ms; TE = 6 ms; flip angle = 15°; FOV = 3.5 cm × 8 cm; matrix size = 128 × 512 × 16; slice thickness = 1.5 mm. Analysis of the change in signal intensity in muscle, heart, liver, kidney and bladder was performed using Matlab. MRI at 7 Tesla was performed on a horizontal Bruker scanner. $T_1$-weighted spin-echo sequence with the following parameters was used: FOV: 3 cm; slice thickness: 1.2 mm; interslice distance: 1.2 mm; TR: 500 ms, TE: 8.1 ms; flip angle: 90°; average: 2; matrix size: 128 × 128. The images were similarly analysed as described above.

**Biodistribution.** The mice injected with the contrast agents were sacrificed 1 week post-injection. Tissue samples were collected, weighed and digested by 1 mL ultrapure nitric acid (EMD Millipore, Billerica, MA, USA) for 7 days. The digested sample (0.5 mL) was diluted to 5 mL with ultrapure water (Milli-Q, EMD Millipore). The solution was centrifuged and filtered using a 0.45 μm filter and the concentration of Gd(III) ions was determined using inductively coupled plasma-optical emission spectrometry (ICP-OES) on a 730-ES ICP-OES system (Agilent Technologies). Samples were measured at three different wavelengths for Gd at 336.224, 342.246 and 358.496 nm and the results were averaged across wavelengths. Intensities were evaluated by ICP Expert II v. 2.0.2 software and were related to the concentration by a calibration curve. A standard calibration curve was developed from a blank and seven standards from a stock solution of 1000 ppm Gd in 3% nitric acid (Ricca Chemical Company, Arlington, TX, USA) and diluted with 2% nitric acid.

**In vitro transmetallation assay.** We performed in vitro transmetallation assay according to the protocol reported previously[29]. Briefly, ZD2-Gd₃N@C80 (0.02 mM Gd/mL) was incubated with human serum for 2 h. According to previous reports[36, 37], transmetallation of Gd with $Zn^{2+}$, $Cu^{2+}$ and $Ca^{2+}$ bound in serum proteins occurs rapidly and 2 h is sufficient to study the equilibrium of transmetallation. The mixtures were then centrifuged at 4000 rpm and 25°C for 150 min using CF-10 centrifugal filters (molecular weight cutoff 10 kDa). Metal ions in both the upper reservoir and the filtrates were quantified by ICP-OES. Transmetallation with $Zn^{2+}$, $Cu^{2+}$ and $Ca^{2+}$ bound in serum proteins resulted in increase in free $Zn^{2+}$, $Cu^{2+}$ and $Ca^{2+}$ in the filtrates. The degree of transmetallation of ZD2-Gd₃N@C80 with $Zn^{2+}$, $Cu^{2+}$ and $Ca^{2+}$ ions in serum was evaluated using the percentage of $Zn^{2+}$, $Cu^{2+}$ and $Ca^{2+}$ ions filtered through the membrane, calculated as Transmetallation (%) = (concentration of ions in the filtrates)/(total ion concentrations before filtration) × 100. Human serum was used a control.

**Statistical analysis.** All the experiments were performed in triplicates unless stated otherwise. No estimation of sample sizes was performed. No randomisation or blinding was used in animal studies. Data are represented as mean ± s.e.m. Analysis of differences between two groups was performed using Student's t-test assuming equal variance, and the difference was considered significant if $P < 0.05$.

**Data availability.** Data available on request from the authors.

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

# ARTICLE

34. Menzin, A. W. et al. Identification of oncofetal fibronectin in patients with advanced epithelial ovarian cancer: detection in ascitic fluid and localization to primary sites and metastatic implants. *Cancer* **82**, 152–158 (1998).

35. Lee, G. Y., Kenny, P. A., Lee, E. H. & Bissell, M. J. Three-dimensional culture models of normal and malignant breast epithelial cells. *Nat. Methods* **4**, 359–365 (2007).

36. Tweedle, M. F. Physicochemical properties of gadoteridol and other magnetic resonance contrast agents. *Invest. Radiol.* **27**(Suppl 1): S2–S6 (1992).

37. Laurent, S., Vander Elst, L., Henoumont, C. & Muller, R. N. How to measure the transmetallation of a gadolinium complex. *Contrast Media Mol. Imaging* **5**, 305–308 (2010).

## Acknowledgements

This research was supported in part by the National Institute of Health Grants R01 EB00489 and CA194518. Z.-R.L. is M. Frank Rudy and Margaret Domiter Rudy's Professor of Biomedical Engineering. We thank Dr. Amita M. Vaidya for the final proof reading and editing of the manuscript.

## Author contributions

The study concept and design were conceived by Z.-R.L. Z.H. and X.W. participated in all aspects of the work. S.R. and C.C. participated in cell and animal studies, as well as ICP-OES analysis. W.P.S. assisted on the tumour models and the biological analysis of the molecular target. Z.H. and Z.-R.L. prepared and edited the manuscript.

## Additional information

**Competing interests:** The peptide and related targeted MRI contrast agents and imaging probes are licensed to Molecular Theranostics LLC. for clinical development. Z.H. and Z.-R. L. are the inventors of the patent and on the patent application. Z.-R.L. is one of co-founders of Molecular Theranostics LLC and has ownership interest of the company.

**Change history:** A correction to this article has been published and is linked from the HTML version of this paper.

