## [Peer Review File · Nature Communications]

Reviewers' comments:

Reviewer #1 (Remarks to the Author):

"What are the major claims of the paper? Are they novel and will they be of interest to others in the community and the wider field?"

The manuscript by Han and colleagues was reviewed. The use of targeted molecular peptide conjugates of gadofullerenes as contrast agents in MRI is very promising. On one hand the Gd(III) ions encapsulated in fullerene cage not only prevents release of toxic Gd(III) ions into the body, the improvements in the relaxivities compared to agents used clinically are extremely impressive. Then to add a targeting peptide that may distinguish between an aggressive cancer and a less aggressive cancer adds to the excitement of this field coming closer to clinical realisation.

I found this work to be very interesting. It is very well written. It appears that the authors follow on from their previous work -reference 18 in which they used an impressive approach to identify peptide ligands that bind to extradomain-B fibronectin (EDB-FN), an oncoprotein associated with epithelial to mesenchymal (EMT) and tumour aggressiveness.

A few minor comments/questions in regards to the work. The sequence provided in this work is missing an amino acid that was mentioned in their previous work. This may be an oversight. See page 4 concerning the ZD2 peptide sequence. On page 5, the paragraph that states EDB-FN is a hallmark of EMT should be changed to EDB-FN is a marker for EMT....Figure 1. Legend gray should be grey? Figure 3 legend – figure 3a – this is confusing. What is the control used to normalise?

In my opinion, although it would be stronger by including additional subgroups of breast cancer cell lines, I believe this is very interesting and would be of interests to others in the field. I found myself wanting more information concerning the method of preparing the fullerenes. Another question I had was do the authors have any thoughts on what their ZD2 peptide may represent in vivo. In other words, does the peptide resemble an endogenous ligand?

The work is convincing and as mentioned above it would be strengthened with additional cell lines representing different subtypes and testing existing in vivo models.

"On a more subjective note, do you feel that the paper will influence thinking in the field? "

I think this would contribute to the feel and influence thinking in the field of non-invasive, molecular targeted imaging. This proof of concept is strong and impressive and I wonder what are the barriers to seeing this move to clinical based studies.

"We would also be grateful if you could comment on the appropriateness and validity of any statistical analysis, as well the ability of a researcher to reproduce the work, given the level of detail provided."

A concern was with the normalisation in the RT-PCR analysis.

"To increase the transparency and openness of the reviewing process, we do support our reviewers signing their reports to authors if the reviewers feel comfortable doing so. If, however, you prefer to send an anonymized report we will continue to respect and maintain your anonymity. Referee reports, whether signed or not, are subsequently shared with the other reviewers."

I am okay with this.

Reviewer #2 (Remarks to the Author):

This is an important manuscript that shows for the first time “the first example of effective non-invasive differentiation between breast tumor models of different aggressiveness using contrast enhanced MRI.” The manuscript also shows with limited data the greater stability of the metallofullerene for encapsulation of toxic Gd(III) which has become of great concern for all Gd containing clinical MRI contrast agents. However, this latter point is not conclusive and the higher level of the agent in Figure 4c in the femur (after 7 days) could argue for retention of the metallofullerene agent in the bone as found in other recent clinical studies. In any case, this paper represents a very significant advance in MR imaging and early detection of aggressive cancers and I recommend publication with minor corrections. Minor comments are listed below:

Title: suggested change.....molecular MR imaging.....

Line 34.....change to.... With superior r1 and r2 relaxivities orWith superior 1/T1 and 1/T2 relaxivities

Line 41 and 42.....” In addition, encapsulation of Gd(III) ions in the fullerene cage prevents release of the toxic Gd(III) ions in the body and facilitates complete excretion of the contrast agent.” Comment: not sure this has been proven to be true (see below)

p.3 Lines 56 & 59.....hydrogen instead of protons

p.3 line 77 change to subscripted C80this error occurs in every place in the ms and needs to be corrected.....hydroxylated Gd3N@C80

p.4, line 80.....”preventing the release of toxic

80 free Gd(III) ions into the body.”.....this has not been proven

p.4, line 90 the MALDI m/z does not match the formula structure for Gd3N@C80(OH)18(COOH)4 or even the corrected structure Gd3N@C80(OH)18(CH2CH2 COOH)4. The authors need to comment on this inconsistency.

p. 7, line 149..... “which is 20 times less than the dose of the clinical contrast agent.”.....Unclear which clinical contrast agent....what is the comparison or reference? p. 8, line 188.....not proven that there is no release.....suggest change to.....and potentially no release of free Gd(III)

p. 11, Figure 1a.....sp..... o-dichlorobenzene

p. 12, Figure 2.....need explanation for m/Z values in a) and b).....they do not match the masses for Gd3N@C80(OH)18(CH2CH2 COOH)4. and ZD2-Gd3N@C80

p. 12 Figure caption.....need magnetic field strength (T) with r1 and r2 values

Reviewer #3 (Remarks to the Author):

Zheng Han et al. synthesized a peptide targeted tri-gadolinium nitride metallofullerene, ZD2-Gd3N@C80, for MRI to detect cancer. The dose of the contrast agent for effective molecular MRI was only slightly lower than that of ZD2-Cy5.5 (0.5 $\mu\text{mol/kg}$) in fluorescence imaging. In addition, encapsulation of Gd(III) ions in the fullerene cage prevents release of the toxic Gd(III) ions in the body and facilitates complete excretion of the contrast agent. The research potentially had significant importance in clinical application. The paper presentation needs more work, I think the manuscript at this stage can't be recommended for publication in Nature Communication, and I made some suggestions below.

1. The language of the manuscript requires an extensive correction by a native English speaker or someone proficient in English. There are some misspellings, unnecessary or lacked space, and it is

difficult to understand some sentences because of language.

2. MRI is a powerful clinical imaging modality that provides high-resolution three-dimensional images of soft tissues. However, due to their slow excretion and accumulation in normal tissues, the safety of the contrast agents must be designed and tested strictly. The safety evaluation of ZD2-Gd3N@C80 is too simple in the manuscript. And the long-term toxicity of ZD2-Gd3N@C80 must be measured. Whether the long-term toxicity of Gd contrast agent, such as renal fibrosis, could be prevented?
3. It is important to set up control group using the clinic Gd contrast agents.
4. What is the final form of ZD2-Gd3N@C80 in the body? What is the main structure of metabolites? Is the carbon cage stable?
5. The authors compared the effect of MRI imaging between the highly aggressive and lowly aggressive tumor using ZD2-Gd3N@C80. They choose MDA-MB-231 triple negative breast cancer cell and estrogen receptor (ER)-positive MCF-7 breast cancer cell. Only a pair of cells is not enough.
6. Please show some data to prove the good water solubility of ZD2-Gd3N@C80. Are the ZD2-Gd3N@C80 nanoparticles easily agglomerated in the different solutions, such as water, NS, PBS, complete medium and so on? And different pH? Do the proteins in physiological solution have influence on ZD2-Gd3N@C80 nanoparticles?
7. The particles with the diameter of < 1.8 nm could be excreted by the kidneys, while ones > 3.6 nm could not be excreted. The authors determined that ZD2-Gd3N@C80 had an average diameter of approximately 1 nm by TEM (Figure 2e) and DLS (Figure 2e). However, I think most of ZD2-Gd3N@C80 are larger than 2 nm from the data. And please provide the particle dispersion index when detected the size distribution of ZD2-Gd3N@C80 using DLS.
8. The authors selected the extradomain B (ED-B), a splice variant of fibronectin, which is exclusively expressed in ovaries, uterus, during wound healing and in tumor tissues. The authors should consider whether it is appropriate and specific as the targeting molecule.
9. With the increase of the applied magnetism, the r_1 value will decrease and the r_2 value will have the opposite change. How about the r_1 and r_2 value of ZD2-Gd3N@C80 in a 3 T or 7 T relaxometer?
10. In figure 3D, the fluorescence intensity was greater in the MDA-MB-231 tumors than the MCF-7 tumors because of the targeted property of ZD2. Why the normal tissues and organs of mouse bearing MDA-MB-231 cells is greater than that of MCF-7 cells?
11. Maybe the authors could use ICP-MS to detect the Gd biodistribution after injection. It is more accurate.

We greatly appreciate the detailed and invaluable comments from the reviewers to improve the quality of our manuscript. Based on the suggestions, we have performed several new experiments and made the relevant changes in the revised manuscript. Please see below for the point by point responses to the reviewer comments.

Reviewer #1 (Remarks to the Author):

"What are the major claims of the paper? Are they novel and will they be of interest to others in the community and the wider field?"

The manuscript by Han and colleagues was reviewed. The use of targeted molecular peptide conjugates of gadofullerenes as contrast agents in MRI is very promising. On one hand the Gd(III) ions encapsulated in fullerene cage not only prevents release of toxic Gd(III) ions into the body, the improvements in the relaxivities compared to agents used clinically are extremely impressive. Then to add a targeting peptide that may distinguish between an aggressive cancer and a less aggressive cancer adds to the excitement of this field coming closer to clinical realisation.

I found this work to be very interesting. It is very well written. It appears that the authors follow on from their previous work -reference 18 in which they used an impressive approach to identify peptide ligands that bind to extradomain-B fibronectin (EDB-FN), an oncoprotein associated with epithelial to mesenchymal (EMT) and tumour aggressiveness.

A few minor comments/questions in regards to the work. The sequence provided in this work is missing an amino acid that was mentioned in their previous work. This may be an oversight. See page 4 concerning the ZD2 peptide sequence. On page 5, the paragraph that states EDB-FN is a hallmark of EMT should be changed to EDB-FN is a marker for EMT....Figure 1. Legend gray should be grey? Figure 3 legend – figure 3a – this is confusing. What is the control used to normalise?

We are honored by the strong positive comments about our work from the reviewer.

ZD2 peptide was initially identified and reported as a cyclic peptide CTVRTSADC with two cysteinyl residues for cyclisation of the peptide. These cysteinyl residues are not essential for specific binding. We have demonstrated that the key sequence TVRTSAD has the same EDB-FN binding properties as the cyclic peptide (DOI: 10.1021/acs.bioconjchem.6b00719). Therefore, we used the linear peptide to simplify the synthesis of the agent in this work. The cysteine on the N-terminus was used for conjugation to maleimido-Gd3N@C80 through Michael addition click reaction. For clarification, we have cited the previous work and discussed the use of this linear peptide in the revised manuscript.

“EDB-FN is a hallmark of” has been changed to “EDB-FN is a marker for”. Legend “gray” has been changed to “grey”.

The legend in Figure 3 has been revised. The data are represented as ratio of mRNA level in MDA-MB-231 cells to the mRNA level in MCF-7 cells. The EDB-FN mRNA levels were normalized with the respective β -actin levels for both the cell lines.

In my opinion, although it would be stronger by including additional subgroups of breast cancer cell lines, I believe this is very interesting and would be of interests to others in the field. I found myself wanting more information concerning the method of preparing the fullerenes. Another question I had was do the

authors have any thoughts on what their ZD2 peptide may represent in vivo. In other words, does the peptide resemble an endogenous ligand?

As suggested, we have incorporated studies and data with additional tumor models. The gadofullerene, Gd₃N@C80, is commercially available. The source of the compound has been included in the manuscript. ZD2 peptide was discovered by phage display by selecting affinity ligands to EDB protein. Although we have not found that it resembles an endogenous ligand, we intend to further investigate whether that is the case.

The work is convincing and as mentioned above it would be strengthened with additional cell lines representing different subtypes and testing existing in vivo models.

We have added four more breast cancer cell lines to this study: ZR-75-1 and T47D, which represent low-risk treatable breast cancer cells; Hs578T and BT549, which represent high-risk triple negative breast cancer cells. The expression of EDB-FN in Hs578T and BT549 cells is higher than that in ZR-75-1 and T47D cells. This data is shown in Figure S3. We also performed MRI studies with these models. Low-risk and slow-growing ZR-75-1 and T47D tumors had less contrast enhancement with the targeted agent than the high-risk BT549 and Hs578T xenografts.

"On a more subjective note, do you feel that the paper will influence thinking in the field? "

I think this would contribute to the feel and influence thinking in the field of non-invasive, molecular targeted imaging. This proof of concept is strong and impressive and I wonder what are the barriers to seeing this move to clinical based studies.

We thank the reviewer for the comment. In order to move clinical studies, we need to perform the FDA required safety tests to obtain approval for an IND.

"We would also be grateful if you could comment on the appropriateness and validity of any statistical analysis, as well the ability of a researcher to reproduce the work, given the level of detail provided."

A concern was with the normalisation in the RT-PCR analysis.

The EDB-FN mRNA levels were normalized with the respective β -actin levels for both the cell lines. For comparison, the data are represented as ratio of EDB-FN mRNA level in MDA-MB-231 cells to that in MCF-7 cells. We have revised the legend of Figure 3 for clarity.

"To increase the transparency and openness of the reviewing process, we do support our reviewers signing their reports to authors if the reviewers feel comfortable doing so. If, however, you prefer to send an anonymized report we will continue to respect and maintain your anonymity. Referee reports, whether signed or not, are subsequently shared with the other reviewers."

I am okay with this.

We thank the reviewer for this.

Reviewer #2 (Remarks to the Author):

This is an important manuscript that shows for the first time "the first example of effective non-invasive differentiation between breast tumor models of different aggressiveness using contrast enhanced MRI." The manuscript also shows with limited data the greater stability of the metallofullerene for

encapsulation of toxic Gd(III) which has become of great concern for all Gd containing clinical MRI contrast agents. However, this latter point is not conclusive and the higher level of the agent in Figure 4c in the femur (after 7 days) could argue for retention of the metallofullerene agent in the bone as found in other recent clinical studies. In any case, this paper represents a very significant advance in MR imaging and early detection of aggressive cancers and I recommend publication with minor corrections. Minor comments are listed below:

We thank the reviewer for the positive comment. As for stability and retention of the agent, the concerns are reasonable for all Gd-based contrast agents. As shown in the literature and in our study, there is no release of free Gd(III) ions from gadofullerenes. The tissue retention of the agent in the tissues is at the detection limit of the ICP-OES we used. Although the retention of our agent in the femur appears relatively higher compared to other tissues, the actual levels are extremely low (<0.5% of injected dose of 5 $\mu\text{mol/kg}$). Because Gd ions are encapsulated in the fullerene cages, there is no release of free Gd(III) ions and there should no deposition of insoluble forms of Gd. Therefore, we believe that eventually, the agent will be completely excreted from the body.

Title: suggested change.....molecular MR imaging.....

We have made this change.

Line 34.....change to.... With superior r1 and r2 relaxivities orWith superior 1/T1 and 1/T2 relaxivities

We have made this change.

Line 41 and 42....." In addition, encapsulation of Gd(III) ions in the fullerene cage prevents release of the toxic Gd(III) ions in the body and facilitates complete excretion of the contrast agent." Comment: not sure this has been proven to be true (see below).

Previous reports in the literature and our study have proved this. We have added these citations to the manuscript.

p.3 Lines 56 & 59.....hydrogen instead of protons

"Proton" is more commonly used when discussing magnetization in MRI.

p.3 line 77 change to subscripted C80this error occurs in every place in the ms and needs to be corrected.....hydroxylated Gd3N@C80

We have made this change.

p.4, line 80....."preventing the release of toxic free Gd(III) ions into the body.".....this has not been proven

The improved metal containment by fullerenes has been demonstrated in various publications by others (DOI:10.1021/ja068639b; DOI:10.1021/ja9093617). We have added these citations to the manuscript. We also performed and added transmetallation study to demonstrate the stability of the C80 cage.

p.4, line 90 the MALDI m/z does not match the formula structure for Gd3N@C80(OH)18(COOH)4 or even the corrected structure Gd3N@C80(OH)18(CH2CH2 COOH)4. The authors need to comment on this inconsistency.

More characterization using XPS was performed in addition to mass spectrometry. The numbers are calculated based on both XPS data and mass spectrometry.

p. 7, line 149..... “which is 20 times less than the dose of the clinical contrast agent.”Unclear which clinical contrast agent....what is the comparison or reference?

We have revised the sentence by adding “Gd-DTPA and Gd(HP-DO3A)” for clarification.

p. 8, line 188.....not proven that there is no release.....suggest change to.....and potentially no release of free Gd(III)

Done.

p. 11, Figure 1a.....sp..... o-dichlorobenzene

Done.

p. 12, Figure 2.....need explanation for m/Z values in a) and b).....they do not match the masses for Gd3N@C80(OH)18(CH2CH2 COOH)4. and ZD2-Gd3N@C80

We replaced MALDI with XPS and FTIR. We believe XPS and FTIR are better characterization methods. The numbers are calculated based on both XPS and mass spectrometry.

p. 12 Figure caption.....need magnetic field strength (T) with r1 and r2 values

Done.

Reviewer #3 (Remarks to the Author):

Zheng Han et al. synthesized a peptide targeted tri-gadolinium nitride metallofullerene, ZD2-Gd3N@C80, for MRI to detect cancer. The dose of the contrast agent for effective molecular MRI was only slightly lower than that of ZD2-Cy5.5 (0.5 $\mu\text{mol/kg}$) in fluorescence imaging. In addition, encapsulation of Gd(III) ions in the fullerene cage prevents release of the toxic Gd(III) ions in the body and facilitates complete excretion of the contrast agent. The research potentially had significant importance in clinical application. The paper presentation needs more work, I think the manuscript at this stage can't be recommended for publication in Nature Communication, and I made some suggestions below.

1. The language of the manuscript requires an extensive correction by a native English speaker or someone proficient in English. There are some misspellings, unnecessary or lacked space, and it is difficult to understand some sentences because of language.

We have revised the manuscript with the help of a native English speaker.

2. MRI is a powerful clinical imaging modality that provides high-resolution three-dimensional images of soft tissues. However, due to their slow excretion and accumulation in normal tissues, the safety of the contrast agents must be designed and tested strictly. The safety evaluation of ZD2-Gd3N@C80 is too simple in the manuscript. And the long-term toxicity of ZD2-Gd3N@C80 must be measured. Whether the long-term toxicity of Gd contrast agent, such as renal fibrosis, could be prevented?

We agree that safety tests are critical for clinical development of any MRI contrast agent. The main cause of the toxicity of current clinical GBCAs is related to the release of free Gd(III) ions. Since encapsulation of Gd(III) ions in the fullerene cage prevents release of the toxic Gd(III) ions

in the body, we wanted to validate the feasibility of the targeted agent first in this work, before performing costly and time-consuming FDA-required safety tests for clinical development.

Nephrogenic systemic fibrosis is only associated with a small portion of patients with renal impairment, while new studies have shown that the stable chelates without release of free Gd(III) ions have a good safety profile. We have added new data on *in vitro* transmetallation study of ZD2-Gd₃N@C80 to demonstrate the high stability as well as the absence of release of free Gd(III) ions from the contrast agent. This is consistent with previous studies (DOI:10.1021/ja068639b; DOI:10.1021/ja9093617), which also indicate a good safety profile of gadofullerenes for clinical use.

Currently, we are working with funding agencies to acquire funds to perform comprehensive safety tests of the agent.

3. It is important to set up control group using the clinic Gd contrast agents.

We recently reported the use of clinical Gd contrast agent, ProHance, as a control in imaging of cancer models (DOI: 10.1021/acs.bioconjchem.6b00719). This agent did show the ability for risk-stratification and specific molecular imaging, as reported in many clinical studies. In addition, we think the ZD2-Gd₃N@C80 structure shares no common characteristics with clinical contrast agents. The use of Gd₃N@C80 as a control should be sufficient to validate the merit of using ZD2-Gd₃N@C80 as a targeted contrast agent for cancer molecular MRI.

4. What is the final form of ZD2-Gd₃N@C80 in the body? What is the main structure of metabolites? Is the carbon cage stable?

The C80 fullerene cage of metallofullerenes is highly stable, even against radiation, which has been demonstrated in other studies (DOI:10.1021/ja9093617). The references have been incorporated in the revised manuscript. Therefore, the cages should be intact after metabolism of the peptide.

5. The authors compared the effect of MRI imaging between the highly aggressive and lowly aggressive tumor using ZD2-Gd₃N@C80. They choose MDA-MB-231 triple negative breast cancer cell and estrogen receptor (ER)-positive MCF-7 breast cancer cell. Only a pair of cells is not enough.

We have added characterization and imaging of additional breast cancer cell lines and tumor models.

6. Please show some data to prove the good water solubility of ZD2-Gd₃N@C80. Are the ZD2-Gd₃N@C80 nanoparticles easily agglomerated in the different solutions, such as water, NS, PBS, complete medium and so on? And different pH? Do the proteins in physiological solution have influence on ZD2-Gd₃N@C80 nanoparticles?

ZD2-Gd₃N@C80 appears as a clear brown solution in water (Figure S1). The DLS characterization and TEM images showed no agglomeration. No influence of proteins on the agent was observed.

7. The particles with the diameter of < 1.8 nm could be excreted by the kidneys, while ones > 3.6 nm could not be excreted. The authors determined that ZD2-Gd₃N@C80 had an average diameter of approximately 1 nm by TEM (Figure 2e) and DLS (Figure 2e). However, I think most of ZD2-Gd₃N@C80 are larger than 2 nm from the data. And please provide the particle dispersion index when detected the size distribution of ZD2-Gd₃N@C80 using DLS.

The diameter of the agent is 2.8 nm, based on DLS. We have also added the polydispersity information in the figure legend.

8. *The authors selected the extradomain B (ED-B), a splice variant of fibronectin, which is exclusively expressed in ovaries, uterus, during wound healing and in tumor tissues. The authors should consider whether it is appropriate and specific as the targeting molecule.*

We believe that EDB is an appropriate target for tumors in organs and tissues, including the breast and prostate, other than those mentioned by the reviewer.

9. *With the increase of the applied magnetism, the r_1 value will decrease and the r_2 value will have the opposite change. How about the r_1 and r_2 value of ZD2-Gd3N@C80 in a 3 T or 7 T relaxometer?*

We added relaxivity measurement at 7T, as show in Figure S2.

10. *In figure 3D, the fluorescence intensity was greater in the MDA-MB-231 tumors than the MCF-7 tumors because of the targeted property of ZD2. Why the normal tissues and organs of mouse bearing MDA-MB-231 cells is greater than that of MCF-7 cells?*

We believe this is normal variation among different mice. Actually, the fluorescent intensity in the normal tissues are not consistently in higher in the mice bearing MDA-MB-231 than those of MCF-7 cells. The relatively high signals in the liver and kidneys are because the agent is excreted via these organs.

11. *Maybe the authors could use ICP-MS to detect the Gd biodistribution after injection. It is more accurate.*

We agree that ICP-MS is more accurate than ICP-OES. We will use it in our future studies.

Reviewers' comments:

Reviewer #1 (Remarks to the Author):

This is a revised manuscript by Han and colleagues. The inclusion of additional breast cancer cell lines and the additional inclusion of data characterizing ZD2-Gd3N@C80 has improved this work. The authors have addressed my concerns.

Just a few minor issues.

1. Is it possible to define TNBC (abstract line 40)? suggest inserting (TNBC) in line 37 following triple negative breast tumors.
2. Please define CNR page 11 line 208.
3. Supplemental figure 4, please consider adding more information such as what concentration? why 2 hours? How this helps support other data showing Gd(III) ions retained in fullerene cage and not released. This would be beneficial to non-specialists.

I believe there will be great interest in this work. The authors continue to move this field forward towards clinical realisation by conducting novel and properly controlled experiments and while there is more to do in terms of FDA approval and clinical use, this work is important as it demonstrates the potential for using gadofullerenes as targeted contrast agents for MRI.

Reviewer #2 (Remarks to the Author):

The authors have responded in a favorable fashion to the comments of this reviewer. This is an important communication and should be published asap. I would only suggest adding the following reference which is a very recent review on the metallofullerene agent employed in this study.

Li, T..... Trimetallic Nitride Endohedral Metallofullerenes: Biomedical Applications. Small,13, (8), (2017)

Reviewer #3 (Remarks to the Author):

They have addressed all my questions. The manuscript still needs presentation editing. It's still verbose with long sentences, but that is something that the editors will need to work with the authors.

We greatly appreciate the positive comments from the reviewers about the revised manuscript. Further revisions have been made based on the reviewers' suggestions. The comments from the reviewers have been addressed point-by-point below.

Reviewer #1

This is a revised manuscript by Han and colleagues. The inclusion of additional breast cancer cell lines and the additional inclusion of data characterizing ZD2-Gd3N@C80 has improved this work. The authors have addressed my concerns.

We appreciate the reviewer's comment.

Just a few minor issues.

1. *Is it possible to define TNBC (abstract line 40)? suggest inserting (TNBC) in line 37 following triple negative breast tumors.*

Done.

2. *Please define CNR page 11 line 208.*

Done.

3. *Supplemental figure 4, please consider adding more information such as what concentration? why 2 hours? How this helps support other data showing Gd(III) ions retained in fullerene cage and not released. This would be beneficial to non-specialists.*

The information has been added in the section of in vitro transmetallation assay in the Methods (page 20). The data have been discussed in the paragraph of pages 13 and 14.

I believe there will be great interest in this work. The authors continue to move this field forward towards clinical realisation by conducting novel and properly controlled experiments and while there is more to do in terms of FDA approval and clinical use, this work is important as it demonstrates the potential for using gadofullerenes as targeted contrast agents for MRI.

We thank the reviewer for the encouraging comments.

Reviewer #2 (Remarks to the Author):

The authors have responded in a favorable fashion to the comments of this reviewer. This is an important communication and should be published asap. I would only suggest adding the following reference which is a very recent review on the metallofullerene agent employed in this study.

Li, T..... Trimetallic Nitride Endohedral Metallofullerenes: Biomedical Applications. Small, 13, (8), (2017)

Done.

Reviewer #3 (Remarks to the Author):

They have addressed all my questions. The manuscript still needs presentation editing. It's still verbose with long sentences, but that is something that the editors will need to work with

the authors.

We have changed some of the long sentences.